# *Lactobacillus fermentum* Inhibits the Development of Colitis by Regulating the Intestinal Microbiota and Metabolites to Enhance the Intestinal Barrier and Decrease Inflammatory Responses

**DOI:** 10.3390/ijms26189181

**Published:** 2025-09-19

**Authors:** Xiaoyan You, Pengcheng Shi, Wenjing Liu, Mengyi Zheng, Lixia Jin, Wei Liu

**Affiliations:** 1Henan Engineering Research Center of Food Microbiology, College of Food and Bioengineering, Henan University of Science and Technology, Luoyang 471023, China; xiaoyanyou@haust.edu.cn (X.Y.); 15290945612@163.com (P.S.); 2School of Medical Technology and Information Engineering, Zhejiang Chinese Medical University, Hangzhou 310053, China; 15170246889@163.com (W.L.); mengyizheng02@zcmu.edu.cn (M.Z.); 3Zhejiang Provincial Key Laboratory of Agricultural Microbiomics, Institute of Plant Protection and Microbiology, Zhejiang Academy of Agricultural Sciences, Hangzhou 310021, China

**Keywords:** *Lactobacillus fermentum*, colitis, intestinal microbiota, metabolomics

## Abstract

Ulcerative colitis (UC) is a chronic inflammatory disorder difficult to cure with current treatments. Therefore, preventive interventions for UC are crucial. This research investigated the intervention potential of *Lactobacillus (L.*) *fermentum* S1 derived from a cat in reducing dextran sulfate sodium (DSS)-triggered UC. Through a combination of physiological, microbiological, and metabolomic analyses, we demonstrated that preventive supplementation with *L. fermentum* S1 significantly mitigated DSS-triggered body weight loss, colon shortening, intestinal barrier (IB) dysfunction, and inflammatory reaction. *L. fermentum* S1 modulated gut microbiota (GM) components and elevated short-chain fatty acids generation. Beneficial genera were abundant (*Akkermansia*, *Ligilactobacillus*, *Turicibacter*, and *Prevotella*_9) while suppressing pathogenic taxa (Parabacteroides and Acinetobacter). Furthermore, *L. fermentum* S1 increased the amount of the anti-inflammatory metabolite hecogenin within the intestines. Spearman’s correlation analysis exhibited significant associations between microbial shifts and metabolite profiles, suggesting that microbiota-derived metabolites can mediate their therapeutic effects. These outcomes indicate the potential of *L. fermentum* S1 as a new dietary supplement and provide a theoretical basis for UC prevention.

## 1. Introduction

Ulcerative colitis (UC) is a chronic, recurrent inflammatory bowel disease (IBD) distinguished by persistent mucosal inflammation that typically begins in the rectum and can progress proximally to involve part or all of the colon [1]. It frequently manifests as persistent or recurrent inflammation of the intestinal mucosa, causing intestinal tissue injury and dysfunction. The primary clinical manifestations of this condition include weight loss, experiencing diarrhea, abdominal discomfort, blood in the stool, and impairment of the epithelial intestinal barrier (IB) [2]. Although the exact pathophysiological mechanisms underlying the development of UC are unknown, they are influenced by several factors, such as genetic susceptibility, epithelial barrier defects, environmental factors, and dysregulated immune responses and microbial ecology. All of these factors have a significant function in UC [3,4].

At the beginning of the 20th century, IBD was almost exclusively observed in Europe. However, IBD incidence in Asian populations has gradually increased over the past five decades, with a rapid increase in newly industrialized countries. Consequently, IBD has developed into a global disease [5]. Symptoms in patients with UC can result in reduced quality of life and impaired physical and mental health [6]. Therefore, developing therapeutic interventions is required to prevent symptom occurrence.

Currently, the primary treatments for IBD are surgery and medication. However, surgical treatment frequently results in a high postoperative mortality rate [7]. Conventional drugs for IBD treatment include 5-aminosalicylate, glucocorticoids, and immunosuppressants. However, pharmacologic treatment can result in multiple adverse effects [8]. As a result, it is of significant practical importance to develop novel approaches for preventing and treating UC that are safe and successful with limited side effects.

Probiotics are live microorganisms that, when provided in sufficient quantities, have an advantageous impact on the host [9]. As research progresses, probiotics have demonstrated significant potential as dietary supplements and drug candidates with first-line medication potential. The intake of probiotics has a positive impact on human gut health. Research indicates that probiotics at concentrations of 10^8^–10^9^ CFU/mL influence the gut microbiota (GM) and influence human health. Studies have supported the role of probiotics in effectively alleviating and addressing IBD by restoring the IB, balancing GM, and adjusting intestinal immune function [10]. Lactic acid bacteria (LAB) are probiotics that promote human health and are essential components of fermented foods, which are a part of human daily life [11]. *Lactobacillus* species were approved by the World Health Organization (WHO) in 2006 as safe probiotics for humans [12]. Earlier investigations illustrated that LAB has significant potential in colitis treatment. *Lactobacillus johnsonii* can convert native macrophages into CD206+ macrophages and produce interleukin (IL)-10 through the TLR1/2-STAT3 pathway to ease colitis [13]. Additionally, *Lactobacillus plantarum* strains can mitigate DSS-triggered colitis through alterations in the GM and immune system [14]. Furthermore, *Lactobacillus rhamnosus* may reduce tumoral resistance and systemic adverse effects by diminishing both systemic and intestinal inflammation [15].

Probiotics are strain-specific, and their mechanisms are still being studied [16]. It is reported that *Lactobacillus* (*L*.) *fermentum* demonstrates broad application visions in preventing and treating enteritis through multiple mechanisms, such as anti-inflammatory [17,18], regulation of intestinal microbiota [19] and antioxidant [20]. This study involved screening and identifying a strain of *L. fermentum* from the gastrointestinal tract of a healthy cat, which was then labeled as *L. fermentum* S1. The effects of *L. fermentum* S1 prophylactic supplementation in DSS-triggered acute UC mice were investigated. The impacts of *L. fermentum* S1 on clinical symptoms, IB, inflammatory factor levels, short-chain fatty acids (SCFAs), intestinal bacteria, and the metabolome of the UC mice model were analyzed to investigate its potential mechanism in disease progression. This research provides a detailed understanding of *L. fermentum* S1 as a new potential probiotic to manage UC, and lays the basis for upcoming development and utilization.

## 2. Results

### 2.1. L. fermentum S1 Intervention Improved UC Symptoms

To examine the interventional impacts of *L. fermentum* S1 on DSS-triggered UC, we evaluated the weight alteration, disease activity index (DAI) score change, the length of the mice colon, and spleen index (SPI) during the experimental period to determine whether *L. fermentum* S1 could alleviate the deterioration of UC.

On the 7th day, compared to the control (Con) group, the DSS group mice depicted a significantly reduced body weight and a significantly elevated DAI value (Figure 1B,C). Colon length signified the DSS severity. The SPI reflected the degree of splenic inflammation in individuals (Figure 1D−F). At the final step of the trial, the DSS group of mice had significantly shorter colons and higher SPI than the Con group. The model was effectively created. Compared to the DSS group, these conditions were alleviated after *L. fermentum* S1 intervention, resulting in significant improvement in weight, significant decrease in DAI, along with a significant restoration of colon length. Moreover, SPI was improved in LF group.

### 2.2. L. fermentum S1 Intervention Attenuated Colonic Tissue Damage

To assess the mitigating effect of *L. fermentum* S1 on colonic tissue and structural damage, we performed hematoxylin and eosin (H&E) staining of the mice’s colon. We observed the morphological changes in the colon across each group of mice through histological analysis. The results indicated that the intestinal wall was significantly thickened, the tissue structure was disorganized, goblet cells were absent, crypt disruption occurred, and neutrophils and inflammatory cells infiltrated the mucosa and submucosa in mice exposed to DSS. However, the LF group exhibited a certain degree of enhancement in histopathological injury compared to the DSS group, exhibiting more cupped cells, a notable reduction in inflammatory infiltration, a mostly organized gland arrangement, and an intact fundamental histological structure (Figure 2).

### 2.3. L. fermentum S1 Intervention Enhances Gut Barrier Integrity

To examine the protective impact of *L. fermentum* S1 on the epithelial IB role in inflammatory colon conditions, we employed immunohistochemistry (IHC) to determine the expression of key barrier proteins ZO-1 and occludin, which are tight junction markers, and MUC2, a marker indicative of goblet cell metaplasia, in the colon (Figure 3A). There was a significant reduction in the ZO-1, MUC2, and occludin expression in the colons of mice exposed to DSS, when compared to the Con group, which is in line with the pathological outcomes from the analysis of colon tissue. However, after *L. fermentum* S1 intervention, the ZO-1, occludin, and MUC2 expression was significantly up-regulated (Figure 3B,D). These results suggested that *L. fermentum* S1 could significantly ameliorate IB injury in mice with DSS-triggered enteritis.

### 2.4. L. fermentum S1 Intervention Attenuates Immune Factors

To understand the immunomodulatory impacts of *L. fermentum* S1 in the UC model mice, we sequentially assessed the inflammatory factors IL-1β/6, the anti-inflammatory factor IL-10, and the inflammatory mediator tumor necrosis factor-α (TNF-α) in the serum of the mice. The outcomes illustrated that the concentrations of TNF-α and IL-1β/6 were significantly reduced in the Con group compared to the DSS group. In comparison with the DSS group, mice displayed a significantly decreased IL-1β and TNF-α after *L. fermentum* S1 intervention (Figure 4A,B,D). However, DSS treatment resulted in decreased serum IL-10, which showed a significant reversion by probiotic supplementation (Figure 4C).

### 2.5. L. fermentum S1 Intervention Restores SCFAs

To examine the influence of *L. fermentum* S1 on SCFAs production in UC model mice, we determined the SCFAs levels in each mouse group through GC. The results indicated that the acetic, propionic, butyric, isovaleric, and valeric acids’ levels, and total SCFAs, in the cecal components of mice treated with DSS were significantly reduced compared to the Con group. The reduction in SCFAs was effectively reversed with the addition of *L. fermentum* S1 intervention, resulting in significant recovery of acetic, propionic, butyric, isovaleric, and total SCFA contents (Figure 5A–E). Additionally, valeric acid content was marginally increased in LF group; nevertheless, this was not statistically significant (Figure 5F).

### 2.6. L. fermentum S1 Intervention Ameliorates Gut Dysbiosis

To examine the influence of *L. fermentum* S1 on the GM in mice with DSS-triggered UC, we conducted 16S rRNA high-throughput sequencing of feces to determine the effect of *L. fermentum* S1. The alpha diversity was compared among the three groups via the Abundance-based Coverage Estimator (ACE) and Shannon indices, where the Shannon index accounts for both species richness and evenness, and the ACE index indicates species richness. Following DSS treatment, ACE and Shannon indices displayed a significant reduction in the DSS group compared to the Con group; nevertheless, the LF group depicted a recovery (Figure 6A,B). Principal coordinates (PCoA) analysis highlighted variations in microbial composition between various experimental groups. A clear division in gut microbial structure was observed between Con, DSS, and LF groups. The introduction of *L. fermentum* S1 greatly reduced the disruption caused by DSS to the GM structure in mice (Figure 6C).

Subsequently, our investigation into the composition of the GM in each group was expanded to include analyses at both the phylum and genus levels. At the phylum level, *Bacteroidota* and *Firmicutes* were the primary phyla of GM in all the groups (Figure 6D). Specifically, the Con group exhibited a composition predominantly consisting of *Bacteroidota* (51.2%) and *Firmicutes* (32.3%). In the DSS group, the proportion of *Bacteroidota* rose significantly to 68.9% compared to the Con group, while *Firmicutes* dropped notably to 21.3%. The introduction of *L. fermentum* S1 intervention resulted in a reduction in the proportion of *Bacteroidota* to 54.2% and an increase in the proportion of *Firmicutes* to 32.3% (Figure 6E). At the genus level, there were differences in GM composition among the groups, predominantly consisting of *Unclassified_Muribaculaceae*, *Bacteroides*, *Ileibacterium*, and *Parabacteroides* (Figure 6F). We performed linear discriminant analysis effect size (LEfSe) analysis and LDA to detect significant differential genera among the groups. LEfSe analysis identified that *Turicibacter*, *Akkermansia*, *Ligilactobacillus*, and *Prevotella_9* were enriched genera in the LF group, with increased *Parabacteroides* and *Acinetobacter* in the DSS group (Figure 6G).

### 2.7. L. fermentum S1 Intervention Alters Metabolites

To investigate the metabolic alterations in the colon of mice with UC and determine the impact of *L. fermentum* S1 therapy on these changes, an untargeted metabolomics analysis was conducted via liquid chromatography-mass spectrometry to identify significantly altered metabolites. The metabolome of the mice’s colonic contents was analyzed, identifying 3079 peaks, with 2775 successfully annotated metabolites.

Principal component analysis (PCA) illustrated significantly separated metabolites between groups in the Con, DSS, and LF groups (Figure 7A). Metabolite alterations were identified using stringent criteria, focusing on metabolites with significant differences (ratio ≥ 1, VIP ≥ 1, *p* < 0.05). The analysis illustrated 63 differential metabolites between the Con and DSS groups, comprising 44 increased and 19 decreased metabolites (Figure 7D). In the Con group, methylbiotin, dopamine, and n-2-ethylhexyl bicycloheptenedicarboximide were significantly up-regulated, whereas lavoltidine and pentahydroxy-9, 10, 14-trimethyl-4, 9-cyclo-9, 10-secocholesta-2, 5-diene-1, 11, 22-trione were significantly down-regulated. Administration of *L. fermentum* S1 resulted in significant metabolite changes compared to the DSS group. The colonic contents of UC model mice exhibited 235 differential metabolites, including 127 up-regulated and 108 down-regulated metabolites (Figure 7E). Hecogenin was significantly up-regulated, while R-1 methanandamide phosphate and nipradilol were down-regulated in the LF group.

We compared the metabolites that changed in the DSS group with those that changed in the Con group and those that changed in the LF group with those that changed in the DSS group and identified 18 co-varying metabolites (Figure 7B). To acquire deeper visions into the metabolic variations among the groups, we performed a hierarchical clustering analysis of these 18 metabolites and constructed a heat map (Figure 7C). The analysis revealed that these metabolites exhibited a restoration of levels following *L. fermentum* S1 administration. These metabolites included n-(1-deoxyfructos-1-yl)-isoleucyl-aspartate 21-glucoside, deferiprone, and barbamide (Appendix A).

To explore the potential metabolic pathways of *L. fermentum* S1 treatment of UC in mice, the annotation of all differential metabolites in DSS and LF groups was performed by means of the Kyoto Encyclopedia of Genes and Genomes (KEGG) database. We screened the most relevant metabolic pathways for *L. fermentum* S1 treatment of UC for metabolic pathway analysis (Figure 7F). The analysis revealed up-regulation in pathways, including lysine degradation, metabolism of D-amino acid, arginine and proline, and the unsaturated fatty acids and secondary bile acid biosynthesis. Conversely, pathways, including pyruvate metabolism, valine, leucine, isoleucine biosynthesis, the biosynthesis of various alkaloids, and tryptophan metabolism, were down-regulated.

### 2.8. Relationship Between GM and Metabolomics

We performed a correlation analysis to determine whether there is a correlation between GM and metabolomics. The results indicated that *Marvinbryantia* and *Unclassified_Bacilli* were correlated with n-(1-deoxyfructos-1-yl)- isoleucyl-aspartate, glycitein, r-1 methanandamide phosphate, pentosidine, 2-(cyclohexylmethylidene)-1, 2, 3, 4-tetrahydronaphthalen-1-one, 6-hydroxymelatonin, and JBIR-125. *Turicibacter* exhibited a significant correlation with 6-hydroxymelatonin and saroclazine A (Figure 7G). The correlation between *Turicibacter* and 6-hydroxymelatonin and saroclazine A was significant. Additionally, indole-3-lactic acid, imazamethabenz-methyl, and JBIR-125 were correlated with GM; however, their association was not statistically significant.

## 3. Discussion

Probiotics, especially lactic acid bacteria, have attracted considerable scholarly interest due to their potential role in modulating GM and augmenting immune function. Evidence suggests that probiotics can affect host health through various mechanisms, such as inhibiting pathogenic organisms, enhancing barrier functions, regulating immune responses, and facilitating neurotransmitter production [21]. Furthermore, the applications of probiotics extend beyond human health, playing crucial roles in animal nutrition and well-being. Studies indicate that these microorganisms improve animal health and production efficiency by modulating the balance and activity of GM [22]. Although probiotics exhibit considerable potential for health promotion, their applications encounter several challenges. Critical issues, including strain specificity, stability concerns, and individual variability in efficacy, necessitate further investigation. Our findings revealed that *L. fermentum* S1 intervention significantly restored body weight, reduced DAI scores, normalized colon length, mitigated pathological damage, repaired IB function, reduced inflammation, and restored SCFA levels, alleviating UC symptoms by regulating the intestinal microbiome and metabolites.

UC develops through a complex process involving IB defects, dysregulated immune responses, and GM imbalances, all of which are vital in starting and sustaining inflammation [4]. UC pathogenesis is significantly influenced by defects in the gut barrier. The IB comprises mechanical, chemical, immune, and biological components that maintain intestinal homeostasis [23]. It includes the mucus layer, epithelial cells, GM, and tight junction proteins. Tight junction proteins (ZO-1 and occludin) are key to epithelial barrier construction, epithelial homeostasis, and preservation of barrier integrity and function [24,25]. In patients with UC, intestinal permeability is increased, accompanied by alterations in tight junction proteins [26,27]. Mucins released by goblet cells are vital for keeping the IB and homeostasis, and MUC2 loss is related to severe intestinal dysfunction [28,29]. In the DSS-triggered UC model mice, ZO-1 and occludin are typically down-regulated, leading to elevated intestinal permeability. In our study, the ZO-1 and occludin concentrations in the colonic tissue of UC mice were significantly boosted by *L. fermentum* S1, indicating its potential in restoring the epithelial IB’s integrity by maintaining tight junction function. Furthermore, it significantly increased goblet cells and MUC2 to restore the mucus barrier, demonstrating its ability to repair the IB function. This suggests *L. fermentum* S1’s efficacy in restoring the epithelial IB’s integrity by preserving tight junction functionality.

Moreover, inflammation is the primary symptom in UC, and the immune response dysregulation is the key pathogenic mechanism. Research has suggested that the dysregulation of the production and inhibition of pro-inflammatory cytokines is closely related to the IBD pathogenesis [30]. IL-1β is a principal modulator of the inflammatory response, predominantly released by monocytes and macrophages in numerous tissues responding to bacterial wall lipopolysaccharides [31]. As a pro-inflammatory cytokine with pleiotropic properties, IL-6 is a significant regulator of the body’s defense mechanisms. It has a vital function in keeping the intestinal homeostasis by contributing to the immune-epithelial-bacteria cross-talk and preserving mucosal integrity [32]. IL-10 is a cytokine that regulates the immune system and is crucial for keeping the intestinal immune balance by suppressing the generation of pro-inflammatory cytokines [33]. TNF-α is an inflammatory cytokine synthesized by macrophages and monocytes during acute inflammatory responses. It is important for the stimulation of lymphocytes, the enhancement of immune cell functions, apoptosis, and inflammation [34]. Dysregulation of TNF-α and IL-1β/6/10 drives the inflammatory processes in UC, highlighting them as potential therapeutic targets. In this study, treatment with the *L. fermentum* S1 strain significantly inhibited the IL-1β/6 and TNF-α generation, and promoted the IL-10 generation. The outcomes illustrated that *L. fermentum* S1 exhibited an excellent anti-inflammatory potential.

SCFAs are major metabolites of the GM, including acids like formic, acetic, propionic, butyric, isobutyric, isovaleric, and valeric, which have a vital function in gut health [35]. Administration of SCFAs enhances the expression of both MUC2 and other mucins, thereby strengthening the mucus barrier [36]. Acetic acid regulates redox signaling and is a crucial energy substrate for intestinal cells [37]. Furthermore, butyric acid is recognized as the principal energy source for colonic cells. It is instrumental in eliciting anti-inflammatory responses by suppressing the NF-κB pathway and the pro-inflammatory gene expression [38]. Total SCFAs, butyric acid, acetic acid, propionic acid, and valeric acid levels were significantly lower in individuals with UC compared to healthy subjects [39]. However, introducing an *L. fermentum* S1 intervention significantly counteracted this reduction and restored the levels of acetic, propionic, and butyric acids, along with the overall SCFA content.

Investigations have indicated a link between UC and an imbalance in GM [40,41]. Individuals with IBD frequently exhibit reduced microbial diversity within the intestinal tract and increased pathogenic bacteria [42]. In mice models of UC induced by DSS, a remarkable decrease was displayed in both the diversity and uniformity of the GM. However, following the intervention with *L. fermentum* S1, these effects were reversed, with a significant restoration of community richness observed in the LF group. Subsequently, we conducted a more in-depth investigation to determine whether *L. fermentum* S1 induces alterations in the GM. The GM predominantly comprises *Bacteroidota* and *Firmicutes* [43]. In cases of UC, there is a significant lessening in the *Firmicutes* to *Bacteroidetes* ratio, which results in decreased biodiversity and ecological dysregulation among patients with UC [44]. Our study restored the ratio of *Bacteroidota* to *Firmicutes* following *L. fermentum* S1 intervention. *Bacteroidota* with *Firmicutes* abundance was restored following *L*. *fermentum* S1 intervention, bringing it closer to the normal group. At the genus level, we observed abnormal proliferation of the potentially pathogenic bacteria *Parabacteroides* and *Acinetobacter* in the DSS group. Previous studies have illustrated that the abundance of *Parabacteroides* can correlate with the severity of UC, as *Parabacteroides* were significantly more abundant during the UC exacerbation than in the period of remission [45]. *Acinetobacter*, as one of the contributing factors to colitis, stimulates the production of inflammatory factors and exacerbates intestinal inflammation [46,47]. Furthermore, we found that the LF group was enriched with beneficial genera, including *Akkermansia*, *Ligilactobacillus*, *Turicibacter,* and *Unclassified_Vicinamibacterales*. *Lactobacilli* are intestinal commensals that help maintain intestinal health and homeostasis [48]. *Lactobacillus* can inhibit inflammatory responses in UC and can ameliorate UC by modulating cytokine profiles and GM [49,50]. *Akkermansia muciniphila* is a commensal microorganism in the human gut that promotes intestinal health by enhancing the intestinal epithelium integrity and the mucus layer thickness. Additionally, it is involved in regulating immune and metabolic functions [51,52]. It exhibits probiotic properties. In addition, *Akkermansia muciniphila* is considered an essential bacterium for propionate production [53]. *Turicibacter* can be a potential probiotic. Research conducted previously indicated that fucoidan raised the relative abundance of the Lachnospiraceae family, including *Turicibacter*, followed by an increase in SCFAs, especially in butyrate [54]. This confirms the capability of *L. fermentum* S1 to encourage the proliferation of helpful bacteria and thereby alleviate UC symptoms. Notably, *Unclassified_Vicinamibacterales* can play a positive role in alleviating colitis as a potential probiotic. In conclusion, *L. fermentum* S1 can be restored by restoring the GM diversity and the phylum abundance, decreasing the pathogenic bacteria abundance, and elevating the number of probiotics.

Metabolites generated by the GM are pivotal in organismal signaling and regulating the immune system, serving as essential mediators of communication between the host and the gut microbiome. Metabolomics analyses exhibited significant variations in metabolite composition between groups, with gavage *L. fermentum* S1 remodeling the metabolic functional pathways of the GM. There were significant alterations in metabolites compared to the DSS group. These metabolites indicated that *L. fermentum* S1 intervention up-regulated lysine degradation, secondary bile acid biosynthesis, d-amino acid metabolism, arginine and proline metabolism, and biosynthesis of unsaturated fatty acids. Additionally, it inhibited the biosynthesis of various alkaloids, pyruvate metabolism, valine, leucine, isoleucine, and tryptophan metabolism. Among them, the metabolite hecogenin exhibits biological activity against numerous diseases both in vivo and in vitro. In addition, it can influence inflammation through various mechanisms. In an in vivo model, hecogenin can suppress IL-1β/6 and TNF-α expression [55].

Furthermore, correlation analysis illustrated correlations between certain differential metabolites and differential flora, with statistically significant results. *Turicibacter* displayed a significant positive correlation with Saroclazine A, and it has been reported that *Turicibacter* is a potential probiotic [56], suggesting that Saroclazine A may exert an anti-inflammatory effect and could serve as a potential therapeutic agent for colitis. In contrast, *Marvinbryantia* displayed a significant positive correlation with N-(1-deoxyfructos-1-yl)-isoleucyl-aspartate. Previous studies have indicated that *Marvinbryantia* acts as a pathogenic bacterium with detrimental effects on gut health, indicating its potential as a pathogen [57]. Consequently, it can be hypothesized that N-(1-deoxyfructosy-l-yl)-isoleucyl-aspartate may exacerbate the condition. These results offer a novel perspective for investigating *L. fermentum* S1 for UC treatment. However, the investigation of the mechanism underlying the intervention of *L. fermentum* S1 in UC is speculative and requires further research.

## 4. Materials and Methods

### 4.1. Preparing L. fermentum S1 Suspensions

*L. fermentum* S1 was obtained from Minsheng Zhongke-Jiayi Biological Engineering Co., Ltd., Hangzhou, China. *L. fermentum* S1 was placed in an MRS liquid medium for cultivation and then spun at 6000 rpm for 5 min after two passages at 37 °C to collect the bacterial cells. The probiotic organisms were then resuspended in phosphate-buffered saline (PBS) to obtain a probiotic suspension of 1 × 10^9^ CFU/mL, which was used for subsequent gastric gavage in mice.

### 4.2. Animal Experiment Design

Male C57BL/6J mice (six weeks old, *n* = 8/group) were brought from Hangzhou Ziyuan Laboratory Animal Technology Co., Ltd. (Hangzhou, China) and housed in a specific pathogen-free facility at the Experimental Animal Centre of Zhejiang Academy of Agricultural Sciences under controlled conditions (25 ± 2 °C, 50% ± 5% humidity, 12 h light/dark cycle and 4 mice per cage) with unrestricted food and water. After seven days of acclimation, mice were classified into 3 groups in a random manner: Con, DSS, and LF (Figure 1A). UC was triggered in DSS and LF groups by administering 3% (wt/vol) DSS in drinking water for a week, while Con mice were administered normal water [58]. The Con and DSS groups were administered 0.2 mL of sterile normal saline (NS) daily through gastric gavage. In contrast, the LF group was administered 0.2 mL of *L. fermentum* S1 (1 × 10^9^ CFU/mL) daily through gastric gavage. The animal ethical approval for this study was granted by the Ethics Committee of the Zhejiang Academy of Agricultural Sciences (no. 2022ZAASLA54).

### 4.3. Sample Collection

Following seven days of treatment, isoflurane inhalation was used to euthanize the mice, followed by cervical dislocation. The collection of blood samples was conducted through eyeball extraction. Fresh blood was collected in tubes without any anticoagulant and spun (4000 rpm, 15 min) to separate the serum, which was then kept at −80 °C until additional analysis. The mice were then euthanized, and feces, colon tissue, colon contents, and the contents of the cecum were gathered for further analysis. Fecal samples were analyzed individually for sequencing.

### 4.4. DAI Scoring

Daily evaluations were made during the trial to demonstrate the existence of stool characteristics, body weight, and occult blood in feces, which were used to calculate DAI [59]. Stool characteristics were as follows: 0 (well-formed pellets), 2 (liquid-like consistency; however, does not adhere to the anus), 4 (adheres to the anus); weight loss: 0 (0%), 1 (1–5%), 2 (5–10%), 3 (10–20%), 4 (>20%); stool occult blood in feces: 0 (no blood), 2 (presence), 4 (bloody stool). We used the SPI to reflect the capacity of immunity. The spleen indices were measured via the formula: SPI = spleen weight/body weight.

### 4.5. Histopathological Analysis

For histopathological analysis, distal colon tissue was excised, followed by fixing in 4% paraformaldehyde, embedding in paraffin, and slicing into 5 µm slices. H&E was utilized to stain the slices for general histological assessment, and inflammation and crypt injury were assessed using a previously established scoring system [13].

### 4.6. IHC Analysis

IHC staining was carried out to analyze the ZO-1, occludin, and MUC2 expression. In brief, colon tissue slices (5 µm) were dewaxed via xylene, rehydrated with ethanol of varying concentrations. Subsequently, the slides were exposed to 3% hydrogen peroxide, and goat serum was utilized to block them at 37 °C for 1 h. The incubation of slides was then conducted with primary antibodies at a dilution of 1:400 at 4 °C. The slides were subsequently incubated in PBS with a biotinylated secondary antibody for an hour at room temperature. Streptavidin-horseradish peroxidase (HRP) was conjugated to the secondary antibody prior to treatment with diaminobenzidine. Lastly, the slides were counterstained with hematoxylin. Microscopic examination of the images was conducted using a Nikon Eclipse C1 microscope (Nikon, Japan), and ImageJ software (Version 1.54) analyzed the intensity, focusing on relative protein expression and average optical density. Hematoxylin gives the nucleus a blue hue, whereas diaminobenzidine’s positive expression is brown-yellow.

### 4.7. Cytokine Assay

An enzyme-linked immunosorbent assay (ELISA) kit was employed to assess the cytokine levels, including IL-1β/6/10 and TNF-α (all from Lunchangshuo Biotech, Xiamen, China) in mouse serum samples, following the manufacturer’s guidelines.

### 4.8. SCFAs Detection

The concentration of SCFAs was quantified using gas chromatography (GC) with a DB-FFAP column (0.32 mm × 30 m × 0.5 µm) and a hydrogen flame ionization detector, using crotonic acid as the internal standard. GC operational conditions were as follows: the inlet temperature was 250 °C, with an injection volume of 1 µL. N_2_ was utilized as the carrier gas, with a purge flow rate of 3.0 mL/L and a split ratio of 8:1 under a pressure of 54.2 kPa. The control mode was linear velocity, with a linear speed of 28.1 cm/s, a total flow rate of 16.1 mL/min, and a column flow rate of 1.46 mL/min. The column oven temperature program started at an initial temperature of 80 °C, with a subsequent increase of 10 °C/min, holding for 0.5 min after reaching 190 °C. The temperature was then elevated at 40 °C/min to a final temperature of 230 °C, which was continued for 4 min. The entire procedure lasted 16.5 min, with detection concluded at 17 min. The flame ionization detector parameters included a temperature of 250 °C, with N_2_ as the tail gas at a 30 mL/min flow rate, hydrogen flow at 40 mL/min, and airflow at 400 mL/min.

Sample pretreatment process: The contents of the cecum were diluted 10-fold with PBS, shaken (2 min), spun (12,000 rpm, 2 min), and then 500 µL of supernatant and 100 µL of internal standard solution were mixed thoroughly and acidified in a refrigerator (−30 °C, 24 h). The supernatant was aspirated and filtered through a 0.22 µm aqueous filtration membrane. After thawing, the sample aspiration was conducted with a disposable syringe and filtered through a 0.22 µm aqueous filtration membrane. Then, it was positioned in a centrifuge tube of 1.5 mL, and 150 µL of the filtered solution was added to a rubber-legged cannula of a gas phase sample bottle, which was subsequently capped tightly to exclude bubbles before sampling.

### 4.9. GM Analysis

Total DNA extraction from mouse fecal samples was conducted via the TGuide S96 Magnetic Soil/Stool DNA Kit (Beijing Tiangen Biotech Co., Ltd., Beijing, China) as per the manufacturer’s protocol. The amplification of the V3–V4 hypervariable region of the bacterial 16S rRNA gene was conducted with primers 338F (5′-ACTCCTACGGGAGGCAGCA-3′) and 806R (5′-GGACTACHVGGGTWTCTAAT-3′). The verification of PCR products was conducted on agarose gel, followed by purification using the Omega DNA purification kit (Norcross, GA, USA), and sequencing via paired-end sequencing (2 × 250 bp) on the Illumina NovaSeq 6000 platform. Additionally, constructing and sequencing the 16S amplicon libraries were conducted on the Illumina HiSeq 2500 platform at Biomarker Technologies Co., Ltd. (Beijing, China). Sequence processing, taxonomic assignment, and alpha/beta diversity analyses were conducted using QIIME2 (version 2022.8), with taxonomic classification based on the SILVA database (version 138). Differential abundance was assessed using LEfSe with an LDA score threshold > 4.0 [60].

### 4.10. Metabolomics Analysis

Extraction of colon contents (50 mg) was conducted with 1000 µL of solution (methanol/acetonitrile/water = 2:2:1, internal standard 20 mg/L), followed by vortexing (30 s), grounding (45 Hz, 10 min), sonicating (ice bath, 10 min), and incubating (−20 °C, 1 h). Samples were spun (12,000 rpm, 15 min, 4 °C), and the collection of 500 µL of supernatant was conducted. A vacuum concentrator was utilized to dry the extract, followed by re-dissolving in 160 µL of acetonitrile/water (1:1), vortexing (30 s), and sonication (10 min, ice bath). After centrifugation (12,000 rpm, 15 min, 4 °C), 120 µL of supernatant was placed in an injection vial of 2 mL. In the end, a quality control sample was created by combining 10 µL of each sample.

A Thermo Scientific Ulti3000 HPLC tandem and a Thermo Scientific Orbitrap Exploris 480 high-resolution mass spectrometer (Thermo Fisher Scientific, Waltham, MA, USA) made up the liquid-mass spectrometry system for metabolomics analysis. The Waters Acquity UPLC HSS T3 column (1.8 µm, 2.1 × 100 mm) was utilized with a scanning range of 67–1000 m/z. Electrospray ionization source settings were positive ion 3.5 kV, sweep gas flow 1 arb, negative ion 2.5 kV, sheath gas flow rate: 50 arb, ion transfer tube temp: 325 °C, aux gas flow rate: 10 arb, and vaporizer temp 350 °C.

### 4.11. Statistical Analysis

All samples underwent triplicate analysis, and data are reported as mean ± standard deviation or standard error of mean. Normality was assessed with the Shapiro–Wilk test. When comparing two separate groups of mice data, a two-tailed unpaired t-test was employed for normally distributed data, and the Mann–Whitney rank sum test was used for data that was not normally distributed. In instances where the data were normally distributed but exhibited unequal variances among groups, the Welch correction was applied. For analyses involving more than two independent groups, one-way ANOVA were applied for comparisons. After confirming significant variations by one-way ANOVA, Tukey’s honestly significant difference (HSD) test was utilized to conduct post hoc comparisons between groups, which adjusts *p*-values to account for several comparisons. *p* < 0.05 was regarded as significance.

## 5. Conclusions

*L. fermentum* S1 exhibited significant potential in restoring the IB, controlling the inflammatory response, and regulating the GM and metabolites. These outcomes illustrate that *L. fermentum* S1 has the potential to prevent the occurrence of UC and can be utilized as a novel potential probiotic dietary supplement for preventing UC.

## Figures and Tables

**Figure 1 ijms-26-09181-f001:**
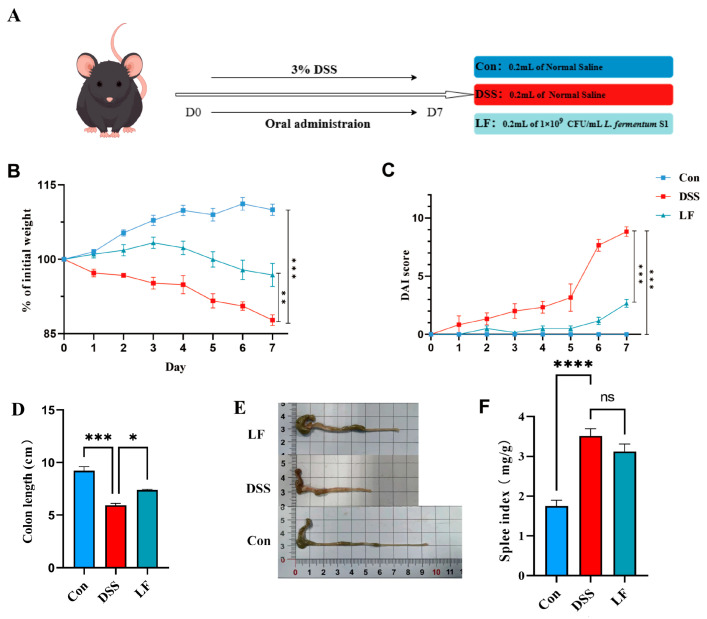
*L. fermentum* S1 enhances the pathological symptoms of the mice with UC: (**A**) Experimental design. (**B**) Percent alterations in the body weights of the mice relative to those on day 1, *n* = 6 mice/group. (**C**) Colon length, (**D**) DAI (*n* = 6 mice/group). (**E**) Images of mouse colon length. (**F**) SPI, *n* = 6 mice/group. *p* < * 0.05, ** 0.01, *** 0.001, **** 0.0001, ns: nonsignificant.

**Figure 2 ijms-26-09181-f002:**
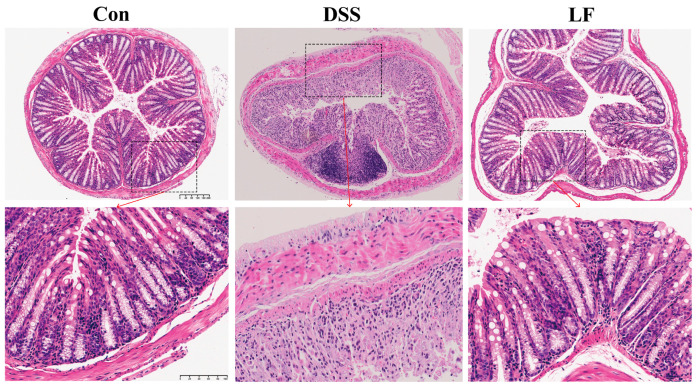
H&E staining of colon tissue slices from all groups. Top shows 10× magnified views, and scale bar is 200 µm. Bottom shows 30× magnified views, and scale bar is 100 µm.

**Figure 3 ijms-26-09181-f003:**
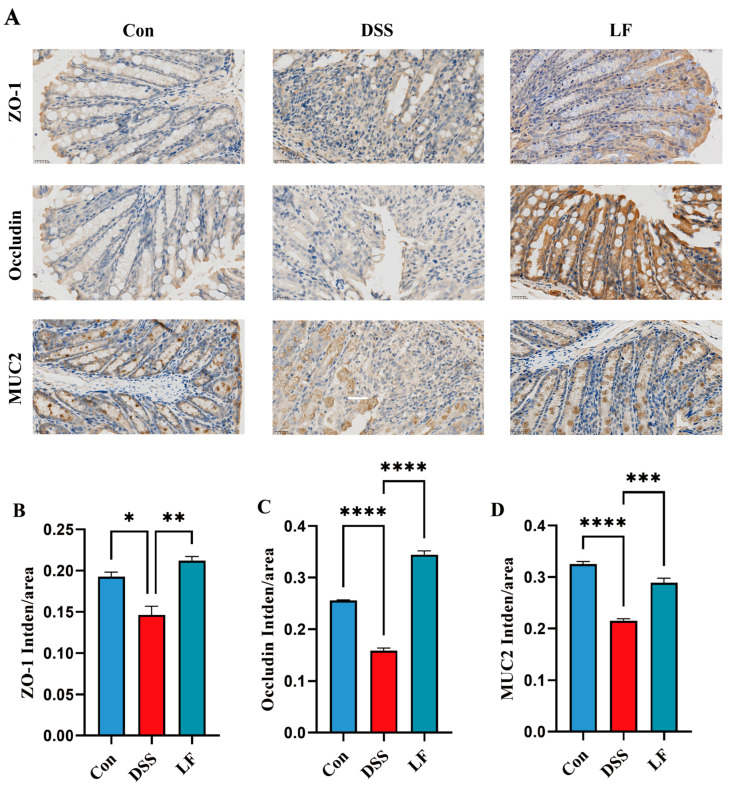
*L. fermentum* S1 ameliorates DSS-triggered IB injury in UC model mice: (**A**) ZO-1, MUC2, and occludin levels in the colon tissues were analyzed through IHC staining (scale bar, 25 µm). The brownish-yellow regions correspond to the target protein, with the average fluorescence intensity analyzed using ImageJ software. (**B**–**D**) Average optical density of ZO-1, occludin, and MUC2 (*n* = 3/group). Data are reported as the mean ± SEM. *p* < * 0.05, ** 0.01, *** 0.001, **** 0.0001.

**Figure 4 ijms-26-09181-f004:**
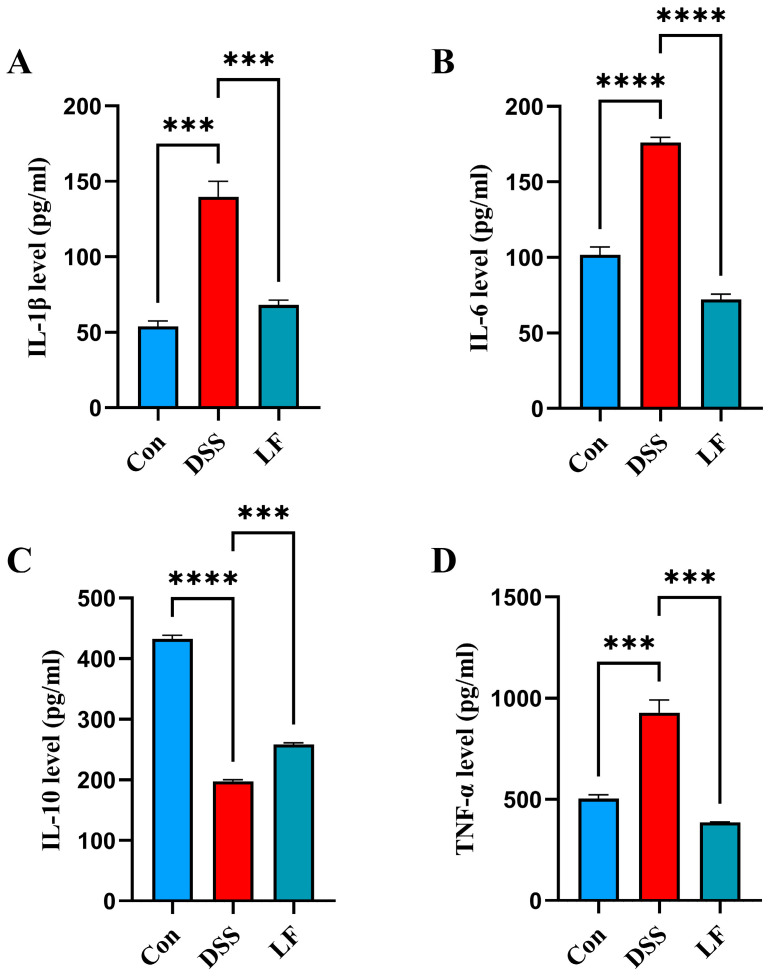
*L. fermentum* S1 improves the colonic inflammation of the DSS-triggered UC model mice: (**A**–**C**) interleukin (IL)-1β/6/10, (**D**) TNF-α (*n* = 3 mice/group). Data are reported as the mean ± SEM. *p* < *** 0.001, **** 0.0001.

**Figure 5 ijms-26-09181-f005:**
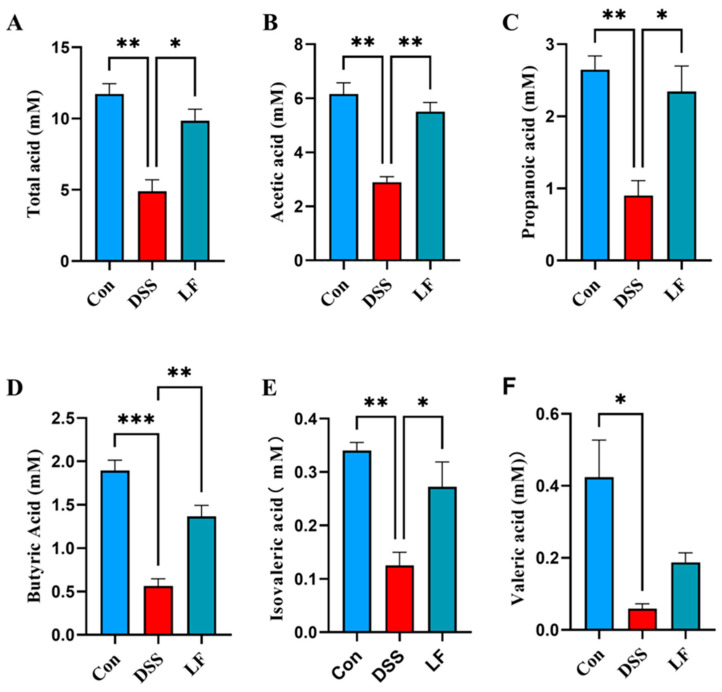
*L. fermentum* S1 improves the SCFAs of the DSS-triggered UC model mice: (**A**) total SCFAs, (**B**) acetic, (**C**) propionic, (**D**) butyric, (**E**) isovaleric, (**F**) valeric acids (*n* = 3 mice/group). Data are demonstrated as the mean ± SEM. *p* < * 0.05, ** 0.01, *** 0.001.

**Figure 6 ijms-26-09181-f006:**
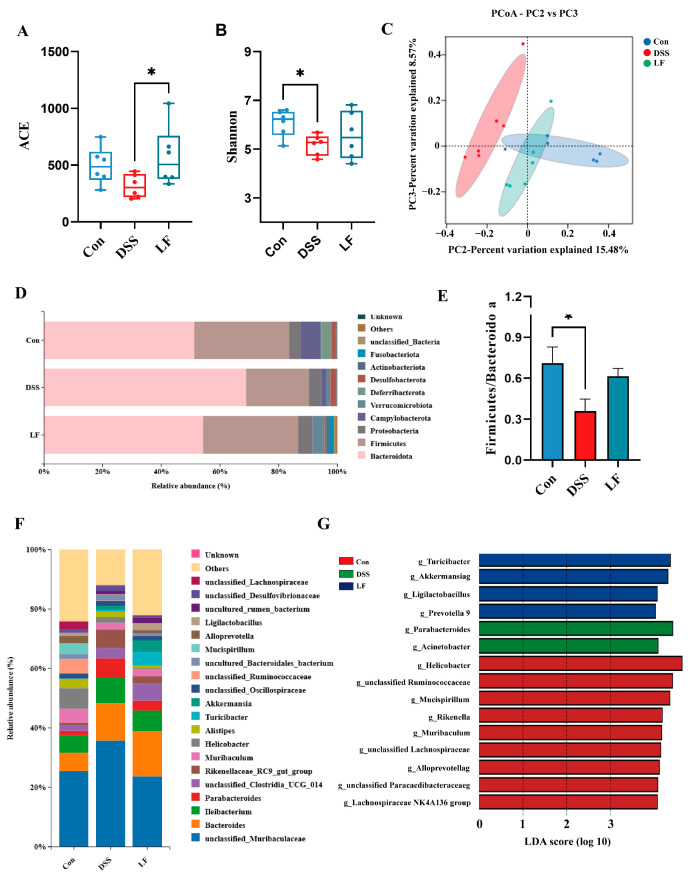
*L. fermentum* S1 regulates GM in UC model mice: (**A**) ACE indexes, (**B**) Shannon indexes, (**C**) PCoA analysis, and (**D**) column charts convey the GM at the phylum level. (**E**) The Firmicutes to Bacteroidota ratio. (**F**) Column charts conveying the GM classified at the genus level. (**G**) LEfSe statistical difference analysis histogram (threshold > 4.0). * *p* < 0.05 (*n* = 6 mice/group).

**Figure 7 ijms-26-09181-f007:**
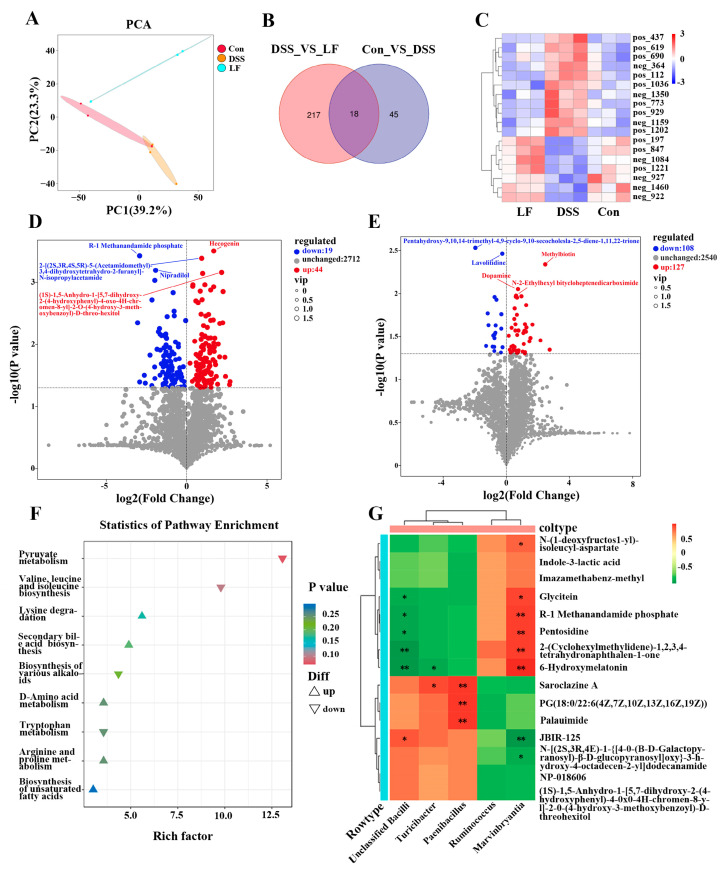
Impacts of *L. fermentum* S1 on colonic contents metabolic profiles in UC model mice: (**A**) PCA. (**B**) A Venn diagram illustrates the two comparisons’ shared metabolites: Con versus DSS and DSS versus LF. (**C**) Related heatmaps of 18 metabolites with common changes. (**D**,**E**) Volcano plots displaying the differential metabolites between the Con versus DSS and the DSS versus LF. Red and blue dots signify increased and decreased metabolites, respectively. (**F**) Metabolite pathway analysis of DSS versus LF. (**G**) The heatmap of Spearman’s correlation between metabolites and microbiota. Different colors represent the value of the correlation coefficient: positive (red) and negative (blue) correlations. *p* < * 0.05, ** 0.01 (*n* = 3 mice/group).

## Data Availability

All 16S rRNA gene sequencing data were submitted to the NCBI with the accession number PRJNA1238789.

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
