# Peer review of "Lactobacillus fermentum Inhibits the Development of Colitis by Regulating the Intestinal Microbiota and Metabolites to Enhance the Intestinal Barrier and Decrease Inflammatory Responses"

_ijms, 2025, doi:10.3390/ijms26189181_

Round 1
Reviewer 1 Report
Comments and Suggestions for Authors
Dear Authors,
I've read you paper with great interest. Please find my comments on the pdf file. I look forward to your revised work.
Best,

Reviewer 2 Report
Comments and Suggestions for Authors
The article examines the protective effects of Lactobacillus fermentum in a model of ulcerative colitis. The study is well designed, and the results are generally well presented and discussed. However, I would like to suggest several points for improvement:
- In the introduction, please provide a rationale for selecting fermentum S1 for this study. In addition, give a brief summary of the previously reported effects of this strain.
- Define the abbreviation DAI when it is first mentioned (line 92).
- Indicate the number of animals included in each analysis within the corresponding figure legends.
- Provide more methodological details regarding ICH staining and quantification. Specifically, clarify which background color was used during staining.
- Line 184: there is an error in the text. Figure 4(E) does not present butyric acid; please correct this.
- Supply higher-resolution images for Figure 6, as the text is currently unreadable.
- Add references and further explanation for the following statements: “Saroclazine A can play an anti-inflammatory role” (line 414) and “Marvinbryantia can have a positive effect on N-(1-deoxysructosyl)-isoleucyl-aspartate, which has significant relevance. Therefore, it can be hypothesized that N-(1-deoxyfructos-1-yl)-isoleucyl-aspartate is involved in exacerbating UC” (lines 419–420). Additionally, ensure that Marvinbryantia is written in italics.
Round 2
Reviewer 1 Report
Comments and Suggestions for Authors
Dear Authors,
I believe that the point by point responses you uploaded are for the other reviewer and not mine. However, I went through the manuscript and found most of the changes and improvements I commented upon. My last request would be regarding the figures as some are still unreadable, for example fig 7 has really small fonts. (also an error = unchageD, the d is missing). You can use the whole page of the manuscript since MDPI does not have any limitations for images, this included the left side of the page (the alignment is restricted only for text not for tables and images). This together with the high DPIs will ensure a better readability of your figures)
Best,
